# Reply to Crawford et al.: Why Trap-Neuter-Return (TNR) Is an Ethical Solution for Stray Cat Management

**DOI:** 10.3390/ani9090689

**Published:** 2019-09-16

**Authors:** Peter J. Wolf, Jacquie Rand, Helen Swarbrick, Daniel D. Spehar, Jade Norris

**Affiliations:** 1Best Friends Animal Society, 5001 Angel Canyon Road, Kanab, UT 84741, USA; 2Australian Pet Welfare Foundation, Kenmore, QLD 4069, Australia; jacquie@petwelfare.org.au; 3School of Veterinary Science, The University of Queensland, Gatton, QLD 4343, Australia; 4Campus Cats NSW, Kensington, NSW 2033, Australia; 5School of Optometry and Vision Science, The University of New South Wales, Sydney, NSW 2052, Australia; 6Independent Researcher, 4758 Ridge Road, #409, Cleveland, OH 44144, USA; 7RSPCA NSW, 201 Rookwood Rd, Yagoona, NSW 2199, Australia; jnorris@rspcansw.org.au

**Keywords:** cat, urban stray, trap-neuter-return, TNR, animal welfare, shelters, infectious disease, euthanasia, conservation, wildlife

## Abstract

**Simple Summary:**

Much controversy surrounds the management of Australia’s unowned urban cat population. The longstanding practice of trapping and killing urban stray cats and kittens that are not reclaimed or adopted has proven ineffective at reducing their numbers. In addition, it has been shown that shelter staff tasked with the repeated killing of healthy animals often face severe mental health consequences. A growing body of scientific evidence from Australia and elsewhere indicates that a non-lethal alternative, trap-neuter-return (TNR), can be effective at reducing urban stray cat numbers. TNR has also been associated with declines in feline intake and euthanasia at animal shelters. A large-scale trial of TNR in Australia is warranted and needed urgently. An extensive yet fundamentally flawed argument against such a trial is presented in a recently published article, “A Case of Letting the Cat out of the Bag—Why Trap-Neuter-Return Is Not an Ethical Solution for Stray Cat (*Felis catus*) Management,” by Crawford et al. In the text that follows, we provided a rebuttal to the Crawford et al. paper and argued that TNR is an ethical, scientifically sound solution for the management of Australia’s urban stray cats.

**Abstract:**

The recently published article, ‘A Case of Letting the Cat out of the Bag—Why Trap-Neuter-Return Is Not an Ethical Solution for Stray Cat (*Felis catus*) Management,’ by Crawford et al. warrants rebuttal. The case presented in the paper, opposing the initiation of TNR trials in Australia, ignores peer-reviewed evidence which substantiates the effectiveness of TNR at reducing unowned urban cat numbers. In addition, the paper’s authors offer a number of unrealistic recommendations, which are little more than a rebranding of the failed status quo. Urban stray cats have long been considered a problem across Australia. Current practice calls for the trapping and killing of thousands of healthy urban stray cats and kittens each year with no apparent effect on the total population. In contrast, the literature offers numerous examples, including two recent studies in Australia, of reductions in urban stray cat numbers where TNR has been implemented. TNR has also been associated with reduced feline intake and euthanasia at shelters, which improves both animal welfare and the well-being of shelter staff. A large-scale trial of TNR in an urban Australian context is scientifically justified and long overdue.

## 1. Introduction

The article, “A Case of Letting the Cat out of the Bag—Why Trap-Neuter-Return Is Not an Ethical Solution for Stray Cat (*Felis catus*) Management,” by Crawford et al. [1] presents a position of opposition to trap-neuter-return (TNR) that conflicts with the available scientific evidence and wrongly concludes that trialing the effectiveness of TNR should not be attempted in Australia. The position taken in this article disregards scientific evidence showing very clearly that TNR can be effective at reducing stray cat numbers on a practical scale and over the long term.

At first glance, ‘A Case of Letting the Cat Out of the Bag’ appears, on the basis of the paper’s length and hundreds of citations, to be a comprehensive review of the topic. A critical reading, however, reveals a flawed central argument supported by an abundance of factual errors and oversights. Our response provides contextual background for this important animal and human issue and refutes the most obvious misstatements in the paper, including a number of unrealistic recommendations.

The reality is that all key stakeholders, including groups that support or oppose TNR, seek the common goal of sustainably reducing the number of unowned urban stray cats in the long term. Current evidence strongly suggests that when TNR is implemented with sufficient intensity [2,3], it will effectively reduce the numbers of stray cats, and therefore, the potential risks posed by urban stray cat populations raised by Crawford et al. Indeed, ‘A Case of Letting the Cat out of the Bag’ unwittingly provides a useful list of reasons why trialing of large-scale TNR is urgently needed in urban Australia.

## 2. Background

The unowned urban stray cat issue is a longstanding problem across Australia. Current methods of stray cat management mainly involve the trapping and killing of thousands of healthy stray cats and kittens each year by shelter, municipal facility and private veterinary practice staff. The killing of healthy cats has been shown to have severe mental health impacts on personnel undertaking the killing [4,5].

Critically, current cat management methods across Australia, which have been undertaken routinely for many years, simply have not decreased the number of unowned urban stray cats in Australia. For example, despite killing 118,000 out of the 196,000 cats impounded by local government from cities and towns across New South Wales over an 8-year period (2008–2009 to 2015–16; 60% euthanasia rate), there was virtually no change in cat intake (25,000 in 2008–2009 vs. 24,000 in 2015–2016) [6]. Since cat intake into shelters, municipal facilities and animal welfare organizations has not significantly decreased over time, it can be concluded that this high level of killing likely did not reduce the stray cat problem.

On the other hand, strong scientific evidence shows that when implemented with sufficient intensity and combined with adoption efforts (as is common practice), TNR can significantly reduce the number of unowned stray cats in urban areas (for example, [7,8,9,10,11]). This in turn reduces intake into shelters and municipal facilities [12,13,14] and the subsequent killing of healthy cats and kittens, saving staff from mental health trauma. Importantly, the benefits of TNR extend well beyond these outcomes. By reducing the numbers of unowned stray cats, other potential associated risks, such as impacts on wildlife, community nuisance, and disease transmission, are also reduced. Furthermore, reducing cat intake into shelters and municipal facilities frees up resources which would have been used for those cats, but which are now available for other shelter animals, such as cats and dogs requiring medical or behavioral care that shelters can address.

We argue that it is now appropriate, indeed urgent, to trial TNR on a large scale in the Australian context. Our position is consistent with that of the RSPCA, Australia’s pre-eminent animal welfare organization. In its recent report ‘Identifying Best Practice Domestic Cat Management’, RSPCA Australia specifically recommended that “A research study should be conducted to evaluate whether, and under what specific circumstances, a program of trap, desex, adopt or return and support (TDARS) is an appropriate tool for urban cat management under Australian conditions”.

## 3. Misunderstanding of the Purpose and Process of TNR

“Letting the Cat out of the Bag” reveals a flawed understanding of the basic principles of TNR. The authors [1] suggest, for example, that “TNR may increase urban cat densities overall” and estimate that 61,000 “new colonies” would be created in Australia’s urban areas if cats were returned rather than euthanized. However, TNR does not create new colonies: after desexing, cats are returned to their original colonies. Many of the paper’s subsequent arguments against TNR stem from this incorrect premise (i.e., that TNR creates new colonies and thereby increases cat densities in urban areas). In addition, the authors define TNR almost entirely as a desexing campaign, without consideration of its vaccination, adoption, and monitoring components, which enhance its efficacy to reduce cat numbers.

Similarly, the authors appear to fall into a common trap of losing sight of ‘baseline conditions’, such as any wildlife and public health impacts associated with free-roaming cats prior to TNR. Simply put, the authors fail to recognize that the cats are already present before instigation of TNR efforts. TNR is a *response* to their presence and mitigates their impacts.

The argument is also made that “removing cats for adoption creates the very ‘vacuum effect’ that TNR colonies are supposed to prevent, with regular removals placing colonies in a permanent state of flux”. In fact, most removals for adoption occur at the start of the TNR program and involve kittens [7,11,12,15]. Later removals are usually limited to socialized immigrant cats. As a result, after instigation of TNR, colonies tend to stabilize relatively quickly in terms of size and social dynamics, thereby allowing their ongoing management and minimizing any ‘vacuum effect’. Moreover, limiting access to other food sources (e.g., garbage) in the area can provide additional control over the number of cats congregating.

The purpose of TNR is to reduce the numbers of urban stray cats over time, and there is ample evidence from Australia and overseas to support this. Conspicuously, ‘Letting the Cat Out of the Bag’ fails to mention two Australian TNR studies demonstrating a 30% reduction in cat numbers over 2 years [8], and a 50% reduction over 5 years [9]. Other studies, including some not cited in the article or recently published (see, for example, [10,11,13,14]) demonstrate significant long-term reductions in urban stray cat numbers (and even elimination of some populations), and reductions in shelter intake and killing.

The data presented by Crawford et al. ([1], Table 1) to quantify outcomes from selected TNR programs are misleading. The table contains a number of inaccuracies, including errors of calculation (e.g., overall numerical response for study 5 should be −85%; data excluding adoptions for study 1 should be −58%). In addition, the duration of study 6 was 1 year, not 2 years [16]. And a total of 120 cats joined study 2 [10], which helps to explain how 180 cats out of 75 were neutered in that program.

Furthermore, analysis of the data in this table is misleading, likely as a result of a misunderstanding of the process of TNR. Most importantly, cats that “joined” the managed colonies during each program are simply ignored in calculating success rates. For example, in the UNSW study [9], 53 cats joined the program after its initiation, increasing the total number of cats managed in this program from 69 to 122. That only 15 cats remained at the end of the 9-year program suggests a −88% response rate rather than −78%. It must be understood that “newly arriving” immigrant cats that might temporarily increase colony size were either already present in the area, born of as-yet un-desexed females, or abandoned by others near the colony [15,17]. Their appropriate management, via ongoing monitoring and desexing, is part of the success of a TNR program.

Examples where Table 1 data suggest that TNR was not effective include study 6, in which less than 20% of the estimated cat population was desexed over a 1-year TNR program (due in part to a numerous logistical challenges), and in which numerous logistical challenges tracking methods introduce significant uncertainties in reported results [16]. Moreover, study 7 showed an increase in cat numbers over a 1-year trial because immigrant and abandoned cats were not promptly managed by desexing or removal for adoption [17].

The article proposes that impoundment, socialization/rehabilitation, and subsequent adoption of stray cats represents an alternative model for management of unowned urban cat populations (i.e., the ‘targeted adoption’ approach). Crawford et al. place considerable emphasis on adoption as the key “to minimize euthanasia while humanely reducing the numbers of stray cats rapidly”, but seem to object to adoption as an integral part of TNR programs, suggesting that the adoption of colony cats overstates TNR’s effectiveness. This attitude is clearly seen in Figure 1, in which success rates from Table 1 are artificially reduced by deleting adopted cats from the analysis, failing to recognise that adoption of colony cats is part of the success of a TNR program.

The authors also warn that “the permanence of adoption and eventual fate of these TNR cats is unknown”. The same can be said for *all* adoptions, of course, yet no similar concern is expressed for the fate of cats adopted through a ‘targeted adoption’ approach. The same bias is displayed in the warning that TNR caregivers might one day find themselves unable to provide necessary care for colony cats, while failing to acknowledge that adopters, too, can face such difficult circumstances.

Referring to a study documenting nine years of a TNR program at the University of New South Wales [9], Crawford et al. highlighted the number of cats that were euthanized, died, or disappeared, arguing that “the manner and number of deaths surely refutes claims about improved cat welfare in the colony”. However, in addition to misrepresenting the numbers of cats “saved … from euthanasia”, there is a failure to provide important context. For example, six of the 21 cats euthanized (29%) were estimated to be more than 10 years old; only one cat (over a 9-year period) was euthanized because of excessively aggressive behavior [9]. Moreover, as Swarbrick and Rand point out, the average mortality rate based on deaths and euthanasia in their campus cat population (including adults and kittens) of approximately 8.1% per annum is consistent with the reported death rate for pet cats of 8.3% per annum [18].

TNR’s demonstrated effect in mitigating ‘shelter overload’ is dismissed in the ‘Letting the Cat Out of the Bag’, even as a study documenting its success is cited as evidence. Following implementation of a TNR program (supplemented by adoption of socialized cats and kittens), Levy et al. [12] reported a 66% decrease in intake of cats and a 95% decrease in the number of cats killed over two years at a Florida shelter. A more recent US study reported similar results arising from 3-year TNR programs: median reductions of 32% in feline intake and 83% in the number of cats killed at six shelters [14]. Such results provide compelling evidence of the positive impact that TNR programs can have on shelters (and their staff). By contrast, the ‘targeted adoption’ approach proposed in the article is more likely to increase feline intake, worsening conditions in shelters already operating at or beyond capacity and, ultimately, resulting in more killing [19,20].

## 4. Alleged Public Health Risks

Crawford et al. argue that the risks of zoonotic disease, nuisance complaints, injuries to people, and allergies “could preclude approval of TNR management”. But again, it is important to acknowledge that these cats already exist in communities across Australia, and that if TNR is implemented with adequate intensity, it will reduce these potential risks by effectively reducing stray cat numbers. In addition, TNR programs typically have an outreach component connecting shelter staff and volunteers with the public, and this disseminates information regarding best practice in community cat management including minimizing legitimate public health risks.

Moreover, ‘Letting the Cat Out of the Bag’ relies heavily on data collected from outside Australia regarding zoonotic risk. This is of limited value given Australia’s geographical isolation and absence of zoonotic diseases such as plague and rabies (which is mentioned extensively despite Australia being rabies-free). In addition, the article suggests that TNR colony cats pose a significant risk for transmitting zoonotic diseases but fails to acknowledge that such transmission generally requires close contact with humans. Although TNR colony cats may be less likely to receive the same level of veterinary care (including vaccination) as pet cats, they typically have limited contact with—and therefore pose relatively low risk to—humans.

Concerns are raised in the article regarding the potential role of cats as vectors for a pandemic. But only one reference is cited to back up this alarming concern, and that research actually studied avian flu (H5N1) transmission between chickens and cats (not humans) under experimental conditions [21].

‘Letting the Cat Out of the Bag’ fails to acknowledge one of the most serious public health risks associated with current urban stray cat management policies in Australia: the impact on the mental health of shelter staff tasked with euthanizing healthy animals [4,5]. Results of a recently published US study reveal a suicide rate among veterinarians that is up to 3.5 times higher than the national suicide rate, with ‘euthanasia procedures’ identified as a likely contributing factor [22]. Staff forced to euthanize healthy cats and kittens often suffer from debilitating ‘moral stress’ [23] and are at high risk of depression and perpetrator-associated traumatic stress. The proposed ‘targeted adoption’ approach would increase shelter intake and euthanasia rates and therefore, exacerbate the public health risk, not reduce it.

## 5. Critical Cost Comparisons Are Missing

Although there is repeated reference to the costs associated with TNR (e.g., “labor and resources to maintain colonies for many years are needed”), the article offers nothing by way of comparison. Data from the US, while admittedly limited, suggest that TNR can provide a considerable cost savings over impoundment and lethal injection: $56 vs. $139 in one Florida shelter, for example [24], and $65 vs. $168 in another [25]. In Cook County, Illinois, it was estimated that taxpayers saved $1.5 million during the first six years of that community’s TNR program [26]. Recent estimates compiled from across the US confirm these figures: $20–97/cat for desexing/return compared to $52–123/cat for impoundment/lethal injection. The cost associated with adoption—of particular interest in the context of the ‘targeted adoption’ approach proposed in the article—was significantly higher: $104–550/cat (Alliance for Contraception in Cats and Dogs, unpublished data).

Higher costs are, not surprisingly, associated with longer lengths of shelter stay. It is estimated that the cost to house and rehome a cat from an Australian shelter is at least $800–1000 AU, given that the average length of stay is 30 days [27,28]. The proposed ‘targeted adoption’ program, which involves “rehabilitation” of unsocialized cats, would very likely result in much greater costs (as well as overcrowded shelters and an increase in the number of cats killed because of space limitations). The authors acknowledge the “clear need for economic research on the relative costs and effectiveness of different proposed strategies for reducing numbers of stray cats in Australian cities” even as they appear to accept, uncritically, the results of a dubious model based on a “single super colony [of 30,000 cats]” [29]. We too welcome additional economic analysis; however, the available evidence strongly suggests that the costs associated with expanded adoption efforts will likely exceed the cost of TNR.

## 6. Misinformation Regarding Impacts on Urban Wildlife Populations

With respect to impacts on wildlife, it is essential to consider ‘baseline conditions’, and to recognize that stray cats are already in the environment prior to instigation of TNR programs. Any intervention that reduces the numbers of urban stray cats, such as TNR, will inevitably also result in a reduction in wildlife predation, an obvious conclusion that is not acknowledged in ‘Letting the Cat Out of the Bag’. Instead, the authors argue that “In Australia, the impact that even small TNR colonies could have on endemic or range-restricted fauna therefore cannot be discounted. Trials of TNR to determine predation impact on these species should not be risked”. Again, the cats are already out there; any potential risks of a TNR trial are, we argue, outweighed by the numerous benefits.

‘Letting the Cat out of the Bag’ also fails to explain that there are no definitive data demonstrating population-level impacts of cats on urban wildlife. Indeed, papers co-authored by one of the authors of ‘Letting the Cat out of the Bag’ include statements such as, “We await definitive experiments that will tell us if predation by pet cats reduces wildlife populations”, “In large urban centres the prey are mainly introduced species such as house mice, starlings and sparrows, but rural cats or those with access to native bushland catch more native species”, and “Despite evidence that pet cats prey on urban wildlife and may transmit disease, there is uncertainty over whether they cause declines in wildlife populations” [30,31].

This latter paper continues: “We suggest that the precautionary principle could be used in this context. The principle mandates action to protect the environment when there is a scientifically plausible but unproven risk” [31]. Importantly, however, a recent paper by Lynn et al. [32] concluded that “it is important to understand that precaution is not simply a rationale for action in the face of scientific uncertainty … The principle of precaution is not a way to sidestep complex questions of science or ethics and thereby resolve the debate over cats one way or the other. It is instead a powerful tool for thinking through and weighing how one ought to respond to cats in varying ecological and social circumstances in light of the ethical and scientific complexities at hand”.

## 7. Other Inconsistencies and Misinformation

A significant amount of the research cited in ‘Letting the Cat out of the Bag’ is misrepresented, often as a result of what has been called “daisy-chaining”, the practice of citing authors who are “merely repeating [material] from what an earlier publication cited” [33]. For example, the claim that “human exposure to rabies is more commonly caused by cats than other domestic animals” fails to acknowledge that the underlying data were compiled during an outbreak of the raccoon variant in the Mid-Atlantic region of the US. And as one of the original sources [34] reveals, the pattern is reversed in other parts of the country, with dog exposures outnumbering cat exposures, as has been documented elsewhere [35]. Similarly, the claim that “Jessup [36] documents studies from the USA that note reduced populations of native bird species, including complete absence of ground foraging species, near sites where unowned cats were fed” is contradicted by the results documented in the original sources [37,38]. In addition, a review of the sources provided to support the claim that “in some studies, poor physical conditions of TNR cats are easily recognizable” [39,40] revealed no such evidence; the stray cats observed were not under TNR management.

More curious is one of the citations used to support a claim that feline immunodeficiency virus (FIV) “is commonly reported in stray cats.” Luria et al. [41] actually reported “similar or lower prevalence rates of infections [in cats admitted to a TNR program] than those published for pet cats” in the US. This is consistent with other research from the US [42] and Canada [43] that documented seroprevalence rates of FIV and feline leukemia virus (FeLV) in TNR colony cats that were comparable to those of indoor-outdoor pet cats. Seroprevalence rates were also lower in cats admitted to Australian shelters than in pet cats with outdoor access [44]. In contrast, significantly higher rates of FIV and FeLV have been observed where no active TNR program had been implemented [45]. The article also fails to properly explain that many cats with FIV remain healthy and do not have any symptoms for years. In fact, many infected cats never develop FIV-related clinical signs and instead eventually die from other causes unrelated to their FIV infection [46].

Also curious is what is missing from ‘Letting the Cat Out of the Bag’. It is noted, for example, that *Toxoplasma gondii* has “been detected in stray, pet and feral cat populations in Australia” and the article warns that “implementing TNR programs may facilitate proliferation of Toxoplasma”. This statement fails to consider, however, that older cats are more likely than younger cats and kittens to be immune as a result of previous exposure to the parasite and thus less likely to shed oocysts [47,48,49]. Even if recently infected, cats older than one year tend to shed fewer oocysts than younger cats [50]. And the likelihood of reshedding oocysts is reduced in cats with good health and nutrition [51]. Thus, Toxoplasmosis risk can be reduced by desexing cats and reducing kitten birth rates, leaving a population of older (and generally healthier) stray cats at a lower risk of spreading toxoplasmosis. Moreover, research has shown that cats living in close proximity to humans—and largely reliant upon human provisions—are much less likely to be exposed to the parasite than “solitary, feral domestic cats living in undeveloped landscapes” [52]. Therefore, deeding desexed stray cats in urban colonies would seem to be an effective measure to reduce the spread of toxoplasmosis in cats, humans, and wildlife.

We agree with Crawford et al. that “capturing, transporting, neutering, vaccinating, worming and medicating are stressful procedures even for well-socialized pet cats, let alone for stray cats unsocialized/partially socialized to human contact”. However, the article fails to acknowledge that the authors’ alternate method would expose cats to the same stresses and more. Cats destined for socialization/rehabilitation experience an average length of stay in shelter confinement of at least 30 days (under current practice in Australian shelters [27]). Under the best practice guidelines, cats that may be euthanized for behavioral reasons should also be kept under observation for at least 3 days to assess the degree to which they might become socialized [53]. An increased length of stay is associated with an increased risk of infectious disease and increases the risk of euthanasia [20,54]. Yet the relationship between intake, length of stay, stress, infectious disease and euthanasia is not considered, nor the potential stress-related consequences of the proposed “targeted adoption” approach.

## 8. Comments on Recommendations

As a result of a lack of understanding of the complex issues around urban stray cat management, significant flaws are made in the article’s recommendations. For example, “targeted adoption” is proposed as an approach to solving the problem. This is despite the fact that evidence already shows that this approach would at best have little impact on stray cat populations, and at worst, is likely to have negative and potentially disastrous consequences by increasing shelter intake and euthanasia rates.

In view of recent efforts by the RSPCA to promote the welfare of impounded cats by reducing euthanasia rates and increasing adoption rates, RSPCA data [55] reveal a telling story. In Queensland for example, between 2011–2012 and 2017–2018 euthanasia rates of impounded cats dropped from 42% to 14%, whereas adoption rates increased from 46% to 69%. However, cat intake did not decrease (13,600 in 2011–2012 vs. 14,900 in 2017–2018). A similar story was shown in RSPCA data from other states and territories [55], suggesting that simply increasing adoption rates of impounded cats is ineffective in reducing the urban stray cat population.

The reality is that Australian shelters are currently unable to rehome all of their rehomeable cats. It is likely that the suggested ‘targeted adoption’ approach will lead to further shelter overcrowding, increased infectious disease rates, increased cat stress, higher costs, higher euthanasia rates (in response to space limitations) and increased negative mental health impacts on staff. The ‘targeted adoption’ approach proposed in the article is not a solution but rather a direct expansion of the existing system; it differs little from the ineffective conventional ‘trap, adopt or kill’ approach, except it would be on a larger scale. The proposal to rehabilitate and rehome friendly socialized cats and euthanize cats with unsuitable temperaments fails to acknowledge that this is current standard practice (undertaken routinely for many years despite a lack of results). This approach has been clearly shown to be ineffective at reducing the number of unowned urban stray cats. This is evidenced by stable cat intake into shelters and pounds despite mass killing over many years [6], and despite efforts to increase adoption rates [55].

## 9. Conclusions

One expects, given the paper’s title, a deep discussion of the ethical theories and frameworks surrounding the management of any sentient beings (e.g., anthropocentric vs. zoocentric, virtue vs. utilitarian vs. deontological, etc.). Not only is such a discussion conspicuously absent, the article reveals no interest in exploring the considerable body of literature on the topic. While it is true that the potential “welfare challenges” identified in the article can be useful prompts for a vigorous debate about the ethics of TNR, they are no substitute for the debate itself [56,57]. Simply declaring TNR to be unethical does not make it so.

We argue that it is inappropriate to dismiss TNR on the basis of the information provided in ‘Letting the Cat Out of the Bag’. Moreover, the complaint that “there is still a dearth of robust evidence demonstrating the long-term success of TNR programs in reducing stray cat population numbers” suggests that the examples cited (see, for example, [7,9,10,11,58]) do not meet the authors’ standards for robustness. Yet, there is no evidence presented that ‘targeted adoption’ will “minimize euthanasia while humanely reducing the numbers of stray cats rapidly” or will be any more effective than TNR. No example of successful implementation of ‘targeted adoption’ anywhere in the world is provided, or any scientific evidence to support the efficacy of this approach.

Of serious concern is that the publication of ‘Letting the Cat Out of the Bag’ is likely to fuel the campaign of misinformation being used to undermine TNR efforts in Australia and elsewhere. It is also likely to deepen the divide between the animal welfare and conservation communities, distracting from what should be obvious common ground: an interest in significantly reducing the number of unowned urban stray cats in a sustainable way.

It is unacceptable on animal welfare, human welfare and ethical grounds to continue to knowingly apply the same ineffective methods (i.e., trap and adopt or kill) repeatedly and fail to research alternatives such as TNR, which has been shown to be effective. A large-scale TNR trial must be undertaken in the Australian urban context to finally address the unowned stray cat problem from a scientific, factual and evidence-based position.

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
