# Peer review of "Reply to Crawford et al.: Why Trap-Neuter-Return (TNR) Is an Ethical Solution for Stray Cat Management"

_animals, 2019, doi:10.3390/ani9090689_

Round 1
Reviewer 1 Report
This article was a good response to the paper by Crawford et al, “A case of Letting the Cat out of the Bag – Why Trap-Neuter-Return Is not an ethical Solution for Stray Cat (Felis catus) Management”. Crawford's article refers to ethics, but does not really come to grips with broader ethical issues, such as the role of humans in the spread of cats populations. The response is measured and rightly points out that TNR needs proper assessment rather than dismissal. The response also points out that TNR should be evaluated in urban areas, not that TNR is a solution in every situation.
Author Response
Thank you for your feedback.
Reviewer 2 Report
This is a helpful assessment of the previously published work which lays out some important assumptions, limitations and inaccuracies.
I have specific comments below to improve the balance and usefulness of this piece is as a rebuttal.
Line 25: “humane” is a word that some would argue wouldn’t apply to TNR and could apply to lethal control. “Non-lethal” is also accurate without having arguable connotations.
Line 66: TNR can be effective and humane when done intensively and well. The newly published paper by Boone et al. would be a great add here. It is important throughout that when discussing population level impacts and declines to clearly acknowledge that sterilization of a cat here or there or only ½ of a group of 10 cats isn’t likely to have a relatively rapid or large impact on population size. So TNR doesn’t always reduce populations with time depending on how it is done.
Section 2: I might call this Background and before line 103 add a heading of Evidence or a similar title.
Paragraph starting on line 87: It would also help to clearly state that the “Cat out of the bag” article is likely defining TNR as only trapping, neutering and returning without adoption (and potentially without vaccinations) or monitoring. That miscommunication or misconception is common and leads to unnecessary conflict. Please define what this current manuscript means when TNR is used for clarity here. This is important for several sections of the manuscript including lines 163. It is also true that TNR with adoption doesn’t lead to an overnight decrease in cat populations. Acknowledging that and putting into the context of the time that the ineffective efforts in shelter have been in existence will strengthen the argument that now is the time to do effective TNR and that the timeline for reduction in populations is consistent with the “Cat out of the bag” proposed solution(s).
Line 94: or resources to provide medical or behavioral care for cats who have needs that shelters can address. Please add.
Line 118 and following: Adding some discussion about the importance of removing or controlling additional access to food like garbage or road kill to help control cat populations would be helpful here.
Lines 146-7: This is a place where monitoring is clearly part of the authors’ definition of TNR which wasn’t explicitly stated. Goes to my previous comment. And this is a critical point…if ongoing monitoring of some sort isn’t being done populations can rise again. The authors make this point elsewhere and this would be a place to emphasize that.
Line 197-8: Please also add that TNR provides a venue to connect to the public and provide information to mitigate existing levels of risk from cats.
Line 217-8: I’m not convinced the veterinarians are doing the majority of euthanasias in animal shelters unless there is data to support that in Australia. Otherwise, please modify and make clear that this is an extrapolation.
Line 244-5: This idea of using a population of 30,000 raises the point that models and implementation of TNR can be done at different, and more practical, population sizes. I think bringing out that point in the manuscript where appropriate would strengthen the argument that effective TNR can 1) be done in the real world and 2) will reduce population size.
Line 312: This should read “be immune”. I might consider references Dubey 1995 and add that good nutrition and health decrease the likelihood of reshedding.
Author Response
Thank you for your feedback—our responses are indicated in blue below.
I have specific comments below to improve the balance and usefulness of this piece is as a rebuttal.
Line 25: “humane” is a word that some would argue wouldn’t apply to TNR and could apply to lethal control. “Non-lethal” is also accurate without having arguable connotations.
The manuscript has been revised accordingly (see Line 25).
Line 66: TNR can be effective and humane when done intensively and well. The newly published paper by Boone et al. would be a great add here. It is important throughout that when discussing population level impacts and declines to clearly acknowledge that sterilization of a cat here or there or only ½ of a group of 10 cats isn’t likely to have a relatively rapid or large impact on population size. So TNR doesn’t always reduce populations with time depending on how it is done.
We agree with the reviewer that intensity matters. The suggested revisions have been made, and references by Miller et al and Boone et al have been added. See Lines 66–67 and 89–90.
Section 2: I might call this Background and before line 103 add a heading of Evidence or a similar title.
The titles and numbering of section headings have now been revised throughout.
Paragraph starting on line 87: It would also help to clearly state that the “Cat out of the bag” article is likely defining TNR as only trapping, neutering and returning without adoption (and potentially without vaccinations) or monitoring. That miscommunication or misconception is common and leads to unnecessary conflict. Please define what this current manuscript means when TNR is used for clarity here. This is important for several sections of the manuscript including lines 163. It is also true that TNR with adoption doesn’t lead to an overnight decrease in cat populations. Acknowledging that and putting into the context of the time that the ineffective efforts in shelter have been in existence will strengthen the argument that now is the time to do effective TNR and that the timeline for reduction in populations is consistent with the “Cat out of the bag” proposed solution(s).
The suggested revisions have been made. See Lines 118–120.
Line 94: or resources to provide medical or behavioral care for cats who have needs that shelters can address. Please add.
The suggested revisions have been made. See Lines 100–01.
Line 118 and following: Adding some discussion about the importance of removing or controlling additional access to food like garbage or road kill to help control cat populations would be helpful here.
The suggested revisions have been made. See Lines 132–34.
Lines 146-7: This is a place where monitoring is clearly part of the authors’ definition of TNR which wasn’t explicitly stated. Goes to my previous comment. And this is a critical point…if ongoing monitoring of some sort isn’t being done populations can rise again. The authors make this point elsewhere and this would be a place to emphasize that.
The suggested revisions have been made. See Line 158.
Line 197-8: Please also add that TNR provides a venue to connect to the public and provide information to mitigate existing levels of risk from cats.
The suggested revisions have been made. See Lines 210–12.
Line 217-8: I’m not convinced the veterinarians are doing the majority of euthanasias in animal shelters unless there is data to support that in Australia. Otherwise, please modify and make clear that this is an extrapolation.
We’re not convinced that veterinarians are doing the majority of euthansias in Australia’s shelters either—we’re referring to the impact on shelter staff more generally. See Lines 228–236.
Line 244-5: This idea of using a population of 30,000 raises the point that models and implementation of TNR can be done at different, and more practical, population sizes. I think bringing out that point in the manuscript where appropriate would strengthen the argument that effective TNR can 1) be done in the real world and 2) will reduce population size.
We have cited several real-world examples, and added (in three places) qualifying language regarding “sufficient intensity” (i.e., Lines 66–67, 89–90, and 209). No further revisions have been made.
Line 312: This should read “be immune.” I might consider references Dubey 1995 and add that good nutrition and health decrease the likelihood of reshedding.
The manuscript has been revised as suggested. See Lines 326–328.